# Stakeholder perceptions of bird-window collisions

**Georgia J. Riggs** *, **Omkar Joshi, Scott R. Loss**

Department of Natural Resource Ecology and Management, Oklahoma State University, Stillwater, Oklahoma, United States of America

* georgia.riggs@okstate.edu

## Abstract

Bird-window collisions are a major source of human-caused avian mortality for which many mitigation and prevention options are available. However, because very little research has characterized human perspectives related to this issue, there is limited understanding about the most effective ways to engage the public in collision reduction efforts. To address this research need, we: (1) evaluated how two stakeholder groups, homeowners and conservation practitioners, prioritize potential benefits and obstacles related to bird-window collision management, (2) compared priorities between these groups, and (3) evaluated potential conflicts and collective strength of opinions within groups. We addressed these objectives by merging the strengths, weaknesses, opportunities, and threats (SWOT) and analytic hierarchy process (AHP) survey approaches. Specifically, survey respondents made pairwise comparisons between strengths and weaknesses (respectively, direct outcomes and barriers related to management, such as fewer collisions and increased costs) and opportunities and threats (indirect outcomes and barriers, such as increased bird populations and fewer resources for other building-related expenses). Both homeowners and conservation practitioners ranked strengths and opportunities higher than weaknesses and threats, indicating they have an overall positive perception toward reducing bird-window collisions. However, key obstacles that were identified included costs of management and a lack of policy and guidelines to require or guide management. These results suggest that substantial advances can be made to reduce bird-window collisions because both homeowners and conservation practitioners had positive views, suggesting their receptivity toward collision management measures. However, because of more neutral views and conflicting responses within the homeowner group, results also highlight the importance of targeting homeowners with education materials that provide information about bird-window collisions and solutions that reduce them. Because bird-window collisions are a human-caused phenomenon, such information about human perspectives and priorities will be crucial to addressing this threat and thus benefitting bird populations.

**Data Availability Statement:** All relevant data are within the manuscript and its Supporting Information files.

**Funding:** This research was funded by Oklahoma State University Department of Natural Resource Ecology and Management (https://go.okstate.edu/

undergraduate-academics/majors/natural-resource-ecology-and-management.html) and Hatch Grant funding (grant numbers: OKL02915, OKL03150) from the USDA National Institute of Food and Agriculture (https://nifa.usda.gov/). Funding was obtained by SRL. The funders had no role in study design, data collection and analysis, decision to publish, or preparation of the manuscript.

**Competing interests:** The authors have declared that no competing interests exist.

## Introduction

As earth's human population continues to grow [1], human actions and ways of life increasingly affect wildlife and their habitats, and the many sources of unintended, direct wildlife mortality are a major component of these human impacts [2–4]. Among direct sources of avian mortality, collisions of birds with buildings and their windows are a top global threat. Window collisions cause between 365 and 988 million bird deaths annually in the United States alone [5] and are also a top threat to birds in other countries (e.g., Canada, Mexico, Brazil, Spain, Singapore, South Korea) [6–11]. Birds collide with glass because they are unable to perceive it as a barrier due to its reflective and transparent qualities [12], and because artificial light at night confuses and draws migrating birds near buildings, elevating collision risk [13, 14]. Bird collisions occur at a wide variety of building types; tall buildings such as skyscrapers have higher per-building collision rates, but smaller and far more abundant residential buildings account for higher cumulative mortality despite lower per building collision rates [5, 7].

Many studies have identified factors that lead to spatiotemporal variation in bird-building collisions. Temporal factors include weather, seasonality, migration phenology, and fluctuations in bird abundance [15–17]. Spatial factors include building-related features like amount of glass, building shape, and nearby vegetation [18–20], as well as broader landscape features like surrounding greenspace and urbanization intensity [21]. Research into correlates of bird-window collisions has led to development of recommendations and management approaches that can be used to reduce collisions. Technologies and commercially available products that reduce glass reflection and transparency have been developed, tested, and marketed, and guidelines to make newly constructed buildings bird-friendly (e.g., by reducing amount of glass or using opaque, fritted, or colored glass) have also been summarized [18, 22, 23]. Municipal, state, and federal policy guidelines and regulations to implement such bird-friendly approaches have also been adopted or are under consideration. These include, for example, *Standards for Bird-Safe Buildings* in San Francisco, California, U.S.A [24], *Buildings, Benchmarks, and Beyond* in Minnesota, U.S.A. [25], *Best Practices for Bird-friendly Glass* and *Best Practices for Effective Lighting* in Toronto, Canada [26], and the *Bird Safe Buildings Act of 2021* currently under consideration by the U.S. federal government [27].

Bird-window collisions occur in areas with human infrastructure, and humans regularly encounter the bird carcasses that result. However, although significant resources have gone into designing and testing mitigation approaches to reduce bird-window collisions, and into developing and implementing bird-friendly policies and guidelines, only two studies have evaluated human perceptions and priorities related to these practices. In fact, there is a general lack of human dimensions research for nearly all sources of direct, human-caused bird mortality, including other kinds of bird collisions (e.g., with wind turbines, communication towers, and vehicles; but see studies of wildlife predation by domestic cats) [28, 29]. One of the studies that evaluated human perspectives related to bird-window collisions examined the Canadian public's willingness to pay (WTP) to reduce collisions at their homes [30] and found that WTP was positively associated with homeowner age, income, and interest in birds, among other factors. The other study investigated public perceptions and knowledge about this issue in Costa Rica and concluded that participants were aware of bird-window collisions but not of the large magnitude of the problem [31]. Clarifying how people perceive bird-window collisions, and how much they support mitigation and prevention techniques, is crucial for bird conservation because implementing effective practices generally entails adoption of new products and technologies on buildings, and therefore, requires buy-in from multiple stakeholder groups (e.g., residential homeowners, owners/managers of commercial buildings, building architects, policymakers).

We began to address this major research gap by exploring and quantifying perceptions and priorities related to bird-window collisions among a diverse pool of respondents in North America. Our objectives were to: (1) evaluate how two important stakeholder groups (owners of individual residences, i.e., "homeowners," and conservation practitioners in state, federal, and non-government conservation organizations) perceive and prioritize potential benefits and obstacles related to bird-window collision management, (2) compare priority rankings for benefits and obstacles to management between homeowners and conservation practitioners, and (3) evaluate potential conflicts in priorities within each stakeholder group, as well as the collective strength of group opinions. To address objectives 1 and 2, we merged the strengths, weaknesses, opportunities, and threats (SWOT) and analytical hierarchy process (AHP) analyses; the approach of merging these two analyses is frequently used to quantitatively assess and rank perceived benefits and obstacles related to management actions and decisions [32–35]. To address objective 3, we used Manfredo et al.'s [36] potential for conflict index (PCI) to visualize within-group conflicts and strength of group opinions, information that can lend additional insight into factors potentially limiting progress in managing bird-window collisions.

## Methods

### Study design

This study, the survey distribution strategy, and the survey contents were approved by and comply with the Oklahoma State University Institutional Review Board's (IRB) standards and regulations (approved IRB protocol # IRB-20-202). All survey participants gave consent for participation upon completion of surveys, and data were also analyzed anonymously. To address objectives 1 and 2, we used a combined SWOT-AHP perception analysis approach (i.e., a strengths, weaknesses, opportunities, and threats analysis linked with an analytic hierarchy process analysis). This merged approach is often used to quantify and rank perceptions about major benefits and obstacles related to issues, actions, and decisions of interest, and to compare benefit and obstacle rankings among diverse stakeholder groups, including for issues in conservation and natural resource management like renewable energy, ecotourism, and land management and policy [32–35, 37, 38]. In the SWOT framework [39], there are 4 categories of factors related to the issue, action, or decision under consideration: strengths, weaknesses, opportunities, and threats. Strengths and weaknesses are considered internal to an issue, action, or decision. In our case, strengths are direct, immediate outcomes of implementing bird-window collision management (e.g., fewer bird collisions) and weaknesses are direct barriers or obstacles to implementing management (e.g., the financial cost of management). Opportunities and threats are considered external to an issue, action, or decision. In our case, opportunities are non-immediate and/or secondary outcomes that indirectly result from implementing management (e.g., increased bird populations due to fewer collisions), and threats are barriers that are not directly related to management but that could arise as management is carried out (e.g., with collision management expenses, reduced financial resources for other building management-related costs). We used the SWOT approach to ask surveyed stakeholders to prioritize strengths, weaknesses, opportunities, and threats related to bird-window collision mitigation and prevention (the specific factors used for each of these 4 SWOT categories are under "Survey Questionnaire Details"). The ultimate goal of a SWOT analysis is to determine perceptions of stakeholders to help develop a strategy that optimizes the tradeoff between strengths and weaknesses of various options, while considering both internal and external factors. When used alone, SWOT does not allow quantitative ranking of factors within or across different categories, making it difficult to draw conclusions about perceptions. The AHP, however, is a generalized method to rank decision problems that assumes independence

among options; when combined with SWOT, AHP allows quantitative comparisons of different SWOT factors, which helps determine the relative importance of a decision [39]. As a multi-criteria decision-making tool, AHP assigns relative weights to factors of interest based on 2-way comparisons between factors [40]; this allows objective evaluation of the degree of agreement (or disagreement) between factors.

## Stakeholder groups and strategy to distribute survey questionnaire

Initially, we sought to investigate priorities of four stakeholder groups: architects, homeowners, and conservation practitioners in both government agencies and non-governmental organizations (NGOs). Each of these groups can play a key role in managing bird-window collisions. Architects can help reduce collisions by working from the top down to incorporate mitigation and prevention measures, within policy parameters, into design and construction of new buildings [41, 42]. Homeowners act from the bottom-up as consumers by expressing their values and desires, buying and living in houses, and deciding whether to manage their properties in ways that benefit birds (e.g., feeding birds or applying films/decals to windows to reduce collisions) [42, 43]. Government and NGO conservation practitioners are both knowledgeable about and advocate for wildlife, but these two groups may enact change in different ways. Government (federal, state/provincial, and tribal) practitioners help inform policy development with research and management, and while NGOs can also help inform policy, they typically engage members of the public through activities such as education campaigns, volunteering, and public funding [41, 42].

To recruit respondents from all stakeholder groups (architects, homeowners, government conservation practitioners, and NGO conservation practitioners) and from as broad of a geographic area as possible, we used snowball sampling, a nonprobability sampling method that uses gateway contacts who can take the survey themselves and are asked to forward the survey invitation to relevant contacts in their stakeholder group [44]. For this study, gateway contacts were the authors' personal or professional contacts in each stakeholder group, including 17 architects, 66 homeowners, 36 government practitioners, and 20 NGO conservation practitioners. Most of these contacts lived and worked in the United States (18 U.S. states represented), but Canada was also represented. Recruitment emails were tailored to each stakeholder group and sent from the authors to gateway contacts; these emails contained a brief overview of the project, a request for participation, a link to sign up to take the survey, a link to a recruitment video on YouTube, and a request that respondents share recruitment materials with colleagues [45]. The recruitment video contained a brief overview about the issue of bird-window collisions and the objectives of this research project, as well as a request for participation and to forward the recruitment materials. In addition to using gateway contacts, we also actively recruited respondents using social media platforms, including Facebook and Twitter [46, 47]. Recruitment via Facebook and Twitter included brief posts on the authors' profile pages, which are followed by numerous professional contacts with formal positions in conservation science and management (including government and NGO conservation practitioners), and by nonprofessional contacts that include numerous homeowners. These Facebook and Twitter posts contained information about the project, the recruitment video, a call for participation, a link to sign up to take the survey, and a request to share recruitment materials. Of note, mixed data collection methods involving focus group meetings, web surveys, and email contacts have been commonly adopted in SWOT-AHP based studies [34, 37, 48]. Accordingly, to broaden participation and increase replication of responses from members of the homeowner stakeholder group, we reached out to multiple neighborhood homeowner's associations (HOA) in Stillwater, Oklahoma, USA, the location of the authors' home

institution (Oklahoma State University). We used this approach because we expected that snowball sampling would result in recruitment of relatively few homeowners. Recruitment materials were sent to publicly available email addresses of HOA board member contacts; again, we requested participation in the survey and dissemination of recruitment materials to other HOA board members and neighborhood residents.

## Survey questionnaire details

Using the merged SWOT-AHP approach first entails development of a survey that contains a list of top strengths, weaknesses, opportunities, and threats regarding the issue at hand. These SWOT lists are often developed from a longer list of candidate factors with assistance of subject-matter experts [37]. We created a list of candidate SWOT factors related to bird-window collision management based on our own subject matter expertise, which includes familiarity with the scientific and gray literature on this topic, and years of interactions and collaborations with key stakeholders in federal/state agencies and NGOs. After drafting the initial list of candidate SWOT factors, we asked three external bird-window collision experts to rank them by importance. Expert responses for each candidate factor were counted and weighted based on ranking to create a final SWOT list containing the four top-ranked factors in each category (Table 1).

Following methodology used by similar SWOT studies, we next solicited stakeholder opinions in two rounds of surveys, with each containing multiple pairwise comparisons between SWOT factors using a scale of one to nine [32, 37, 49]. Specifically, a value of 1 indicated an opinion that one factor was "extremely important," a value of 9 indicated an opinion that the other factor was extremely important, and a value of 5 indicated an opinion that the two factors were "equally important" (see Fig 1 for visual representation of scale). For Survey 1, all possible pairwise comparisons were made between factors *within* (but not between) SWOT categories. For example, all possible 2-way comparisons were made among strengths (e.g., *Fewer collisions* compared to *Fewer bird carcasses to clean up*), but in this survey, strengths were not compared to weaknesses, opportunities, or threats (see example comparison in Fig 1 and S1 File for full Survey 1 contents). We created Survey 2 based on top-ranked factors

**Table 1. List of all SWOT factors.**

| Strengths | Weaknesses |
|---|---|
| S1: Fewer collisions | W1: No economic incentives building for bird-friendly buildings |
| S2: Fewer carcasses to clean up | W2: Lack of architect experience in bird-friendly design |
| S3: Fewer people witnessing collisions | W3: Lack of availability of expert consultation for bird-friendly design |
| S4: Fewer stunned birds that die of other causes while recovering from colliding | W4: Financial burden of treating glass or including bird-friendly design in building process |
| **Opportunities** | **Threats** |
| O1: Recovering bird populations | T1: Unknown social acceptance of bird-friendly treatments and design |
| O2: Public exposure to bird-friendly options | T2: Lack of understanding of federal/state policy on bird-window collisions |
| O3: Consideration of birds in building design becoming a norm/standard | T3: Reduced resources available to spend on other facilities maintenance/improvements |
| O4: Greater energy efficiency of buildings | T4: No federal/state policy in many areas |

Finalized list of strengths, weaknesses, opportunities, and threats (SWOT) containing the top four factors for each category that were used to evaluate stakeholder perceptions regarding bird-window collisions.

**Block 3**

**B1: Strengths**

Please carry out a pairwise comparison of the following set of factors that are likely to be considered Strengths of bird-window collision mitigation and prevention. Please mark the factor you think is more important than the other. For example, compare the factor "Fewer collisions" with "Fewer carcasses to have to clean up" and mark the option in the direction that accurately reflects the degree of your opinion. Please note there is no 'right' or 'wrong' answer, we are simply interested in your opinion.

**Fig 1. SWOT survey example.** Examples of pairwise comparisons within the strengths category of the strengths, weaknesses, opportunities, and threats (SWOT) analysis; this example illustrates the format of Survey 1 distributed to stakeholder groups to evaluate their perceptions and priorities regarding bird-window collision management.

calculated from Survey 1 for each SWOT category (see details of these calculations under "Data Analysis"). These calculations were made separately for each stakeholder group, which allowed us to tailor Survey 2 to each group, a standard practice for SWOT studies. In Survey 2, respondents were asked to make pairwise comparisons of all top-ranking factors *between* SWOT categories. For example, within the homeowner group, the factor *Fewer collisions* was identified as the top strength in Survey 1, and *No federal/state policy in many areas* was the top threat. Thus, respondents were asked to compare these two factors (see S2 and S3 Files for full Survey 2 contents for each stakeholder group).

All surveys were administered using the online platform Qualtrics [50], and both surveys had the same general format. Both surveys contained an introductory page displaying information about the study, including the study's purpose, what to expect, risks associated with participating, and a confidentiality statement. Next, the survey asked respondents to indicate which stakeholder group they belonged to. The following section contained a brief introduction to the issue of bird-window collisions (to give respondents introductory background or to reorient them to the issue), as well as a table containing all of the SWOT factors. To minimize the collection of personally identifiable information and to retain survey anonymity, we only collected contact information (names and emails) of potential respondents during the initial recruitment period (i.e., the period during which we asked stakeholders to sign up to take the survey, but before the survey was distributed). During survey periods, surveys were completed anonymously; therefore, we could not monitor which people who signed up (including gateway contacts and other people reached through snowball and purposive sampling) actually

completed the surveys. For Survey 2, all individuals who signed up to take Survey 1 were again contacted, but we requested that only those that completed Survey 1 complete Survey 2. Survey 1 was administered from 1 June 2020 to 30 June 2020, and Survey 2 was administered from 13 July 2020 to 12 August 2020. For all stakeholder groups and sampling approaches, we waited two weeks before sending one reminder to complete the survey to allow adequate time for participants to respond to the original request [51].

### Data analysis

Analyses of survey response data followed methods of other SWOT-AHP studies (e.g., Starr et al. 2019 and Joshi et al. 2020) [37, 52] that adapted their analyses from Saaty [40]. The same general procedures were used to analyze results from Survey 1 (comparisons within SWOT categories) to determine factor priorities for Survey 2, and to analyze results from Survey 2 (comparisons of top-ranked factors between SWOT categories). First, to calculate the weighted geometric mean for each factor in each SWOT category, and also separately for each stakeholder group, we collated response data for each pairwise comparison into counts according to the selection scale of one to nine (See S1 Dataset for calculated geometric means). Counts were then weighted reciprocally, multiplied, and taken to the power of one over the total number of counts [53]. Each weighted geometric mean was entered into a standard reciprocal matrix, and values were then normalized and placed into a weighted reciprocal pairwise matrix. The weighted reciprocal pairwise matrix was used to calculate factor priority values for each factor in each SWOT category and stakeholder group; these values were used to evaluate relative importance of factors within each SWOT category (all factor priority values within a category sum to one). The standard reciprocal matrix and factor priority values for each category were also used to calculate a consistency index, which when used with a predetermined random index (based on the number of SWOT factors within a category) determines the consistency ratio, a metric indicating the consistency of responses among respondents within a stakeholder group [39, 52]. Pairwise comparisons within each SWOT category were determined to be internally consistent if the consistency ratio (calculated for each SWOT category within each stakeholder group and for both surveys) was less than 10%; however, consistency ratios up to 20% are considered acceptable [34, 40, 49, 52]. When we conducted preliminary analyses of Survey 1 responses, we calculated unacceptably high consistency ratios within the architect and NGO practitioner groups that were most likely attributable to small sample sizes of recruited respondents (n = 12 for each group). We therefore excluded data for architects, and due to similarities between the groups and to prevent data loss, we combined government practitioners (n = 26) and NGO practitioners into a single group (conservation practitioners, n = 38). Thus, our final analysis of Survey 1 (and subsequently, Survey 2) included two stakeholder groups, homeowners (Survey 1: n = 52; Survey 2: n = 33) and conservation practitioners (Survey 1: n = 38; Survey 2: n = 41). Our receipt of more conservation practitioner responses for Survey 2 than Survey 1 was unexpected because we only asked recruits to complete the second survey if they had already completed the first survey. This result likely arose because we had to exclude a small number of Survey 1 responses that were incomplete or contained response errors (7 surveys excluded for homeowners; 4 for conservation practitioners). Regardless of the explanation, we have no reason to believe that receiving slightly more Survey 2 results biased our results.

The last steps in the SWOT-AHP analysis were to calculate global and group priority values. Global priority values rank individual SWOT factors among all categories for each stakeholder group; these values allow for comparison among stakeholders' perceptions and priorities, as well as evaluation of SWOT factor priority rankings against each other [32, 37, 49]. Global priority values within each SWOT category were then added together to create group priority

values that represent the priority of each SWOT category as a whole. We also followed previous literature (e.g., Dwivedi & Alavalapati 2009 and Joshi et al. 2018) [32, 34] to generate perception maps, which illustrate differences in global priority values and allow direct comparisons among all SWOT factors and between stakeholder groups.

To address objective 3, we applied Manfredo et al.'s [36] potential for conflict index (PCI) to the Survey 2 responses (see S1 Dataset for PCI calculations); the PCI allows visualization of potential conflicts in perceptions within stakeholder groups, and of the collective strength (vs. neutrality) of group opinions [54], information that can lend additional insight into factors potentially limiting progress in addressing bird-window collisions. We used the $PCI_2$, an extension of PCI that is used for response data from a scalar survey to visually display degree of conflict (i.e., opposite of agreement) in responses among respondents in a stakeholder group, as well as neutrality of responses [36, 54]. In this case, the scalar survey questions were pairwise comparisons that respondents completed in Survey 2. With regard to neutrality, pairwise comparisons that are near five for a stakeholder group indicate factors perceived as *Equally important* (indicated as bubbles close to the x-axis on PCI graphs). Comparisons that are lower (near one) or higher (near nine) toward either of the factors being compared represent an average group perception that one factor is *Extremely important* relative to the other (bubbles farther from the x-axis). Regarding degree of conflict, this value ranges between 0 and 1, with values close to 0 indicating little conflict (strong agreement on a pairwise comparison among respondents in a group, indicated as small bubbles), and values close to 1 indicating complete conflict (i.e., responses on a pairwise comparison equally divided between the two extreme values on the response scale, indicated as large bubbles) [36, 55].

## Results

### Stakeholder priorities for different SWOT categories

Our survey likely had a nationwide or even broader scope, as our gateway contacts represented at least 18 U.S. states and Canada. However, the exact geographic distribution of survey respondents is unknown because surveys were completed anonymously to minimize collection of personally identifiable information, and because the snowball sampling method we used entailed recruitment of additional respondents beyond our gateway contacts. For all SWOT categories except two in the conservation practitioner group for Survey 1, consistency ratios were <10%, indicating consistent responses within stakeholder groups. For conservation practitioners, the weaknesses and opportunities categories had consistency ratios of 19% and 18%, respectively, indicating some inconsistency. Nonetheless, consistency ratios <20% are considered acceptable for drawing inferences [34, 49].

A summary of SWOT factor, group, and global priorities for homeowners and conservation practitioners is in Table 2. Group priorities for homeowners for strengths, weaknesses, opportunities, and threats were 24%, 15%, 40%, and 21%, respectively, and group priorities for conservation practitioners were 24%, 15%, 52%, and 9%, respectively. For homeowners and conservation practitioners, perceptions about potential outcomes of bird-window collision mitigation and prevention were generally positive, as evidenced by summed percentages of group priorities for strengths and opportunities (64% and 76% for homeowners and conservation practitioners, respectively). As indicated by group priority values for threats, homeowners gave greater priority (21%) to threats than did conservation practitioners (9%).

### Stakeholder priorities for different factors within SWOT categories

As evident from the above-presented group priority values, homeowners prioritized opportunities overwhelmingly over strengths, weaknesses, and threats. Among opportunities,

**Table 2. Factor, global, and group priorities for all SWOT factors for each stakeholder group.**

| SWOT Factors | Factor Priority | | Global Priority | |
|---|---|---|---|---|
| | Homeowner | Conservation Practitioner | Homeowner | Conservation Practitioner |
| S1: Fewer collisions | **0.46** | **0.60** | 0.11 | 0.15 |
| S2: Fewer carcasses to clean up | 0.11 | 0.06 | 0.03 | 0.02 |
| S3: Fewer people witnessing collisions | 0.09 | 0.07 | 0.02 | 0.02 |
| S4: Fewer stunned birds that die of other causes while recovering from colliding | 0.34 | 0.27 | 0.08 | 0.07 |
| Group Priorities for Strengths | | | **0.24** | **0.24** |
| W1: No economic incentives for building for bird-friendly buildings | 0.23 | **0.36** | 0.03 | 0.05 |
| W2: Lack of architect experience in bird-friendly design | 0.18 | 0.13 | 0.03 | 0.02 |
| W3: Lack of availability of expert consultation for bird-friendly design | **0.31** | 0.26 | 0.05 | 0.04 |
| W4: Financial burden of treating glass or including bird-friendly design in building process | 0.28 | 0.25 | 0.04 | 0.04 |
| Group Priorities for Weaknesses | | | **0.15** | **0.15** |
| O1: Recovering bird populations | **0.34** | **0.45** | 0.14 | 0.23 |
| O2: Public exposure to bird-friendly options | 0.18 | 0.15 | 0.07 | 0.08 |
| O3: Consideration of birds in building design becoming a norm/standard | 0.25 | 0.20 | 0.10 | 0.10 |
| O4: Greater energy efficiency of buildings | 0.23 | 0.21 | 0.09 | 0.11 |
| Group Priorities for Opportunities | | | **0.40** | **0.52** |
| T1: Unknown social acceptance of bird-friendly treatments and design | 0.19 | 0.14 | 0.04 | 0.01 |
| T2: Lack of understanding of federal/state policy on bird-window collisions | 0.25 | 0.16 | 0.05 | 0.01 |
| T3: Reduced resources available to spend on other facilities maintenance/improvements | 0.25 | **0.36** | 0.05 | 0.03 |
| T4: No federal/state policy in many areas | **0.31** | 0.35 | 0.07 | 0.03 |
| Group Priorities for Threats | | | **0.21** | **0.09** |

Summary of factors used in strengths, weaknesses, opportunities, and threats (SWOT) analyses related to perceptions and potential outcomes of bird-window collision mitigation and prevention. Factor priority values indicate the relative importance of a single factor within a SWOT category among other factors in the same category (boldfaced factor priority values are the highest prioritized factor for each SWOT category). Global priority values rank individual SWOT factors among all factors and can be compared across SWOT categories. Group priority values (the boldfaced values in "Global Priority" columns) are the sum of global priority values within each SWOT category and are used to compare categories against each other.

*Recovering bird populations* was the top factor priority (34%), followed by *Consideration of birds in building design becoming a norm/standard* (25%) and *Greater energy efficiency of buildings* (23%). Homeowners prioritized strengths next; highest priority strengths were *Fewer collisions* (46%) and *Fewer stunned birds that die of other causes while recovering from colliding* (34%). The anthropocentric strengths received lower priority, including: *Fewer carcasses to clean up* (11%) and *Fewer people witnessing collisions* (9%). For threats, which homeowners prioritized only slightly behind strengths, the top factor was *No federal/state policy in many areas* (31%), followed by two equally ranked (25%) priorities: *Lack of understanding of federal/state policy on bird-window collisions* and *Reduced resources available to spend on other facilities maintenance/improvements*. Homeowners prioritized weaknesses lowest, with *Lack of availability of expert consultation for bird-friendly design* being the top priority (31%) within this category (Table 2).

Based on group priority values, conservation practitioners also prioritized opportunities as most important; among opportunities, *Recovering bird populations* was the top-priority factor (45%). Strengths was the second-highest prioritized category, and top factors in this category were *Fewer collisions* (60%) and *Fewer stunned birds that die of other causes while recovering from colliding* (27%). Conservation practitioners gave weaknesses and threats lowest priority.

The most highly prioritized weakness was *No economic incentives for building bird-friendly buildings* (36%); the two top threats were *Reduced resources available to spend on other facilities maintenance/improvements* (36%) and *No federal/state policy in many areas* (35%) (Table 2).

## Stakeholder priorities for different factors across SWOT categories

Perception maps (Fig 2A and 2B) illustrate differences in global priorities and allow direct comparisons among all SWOT factors and between stakeholder groups. For homeowners, the opportunity *Recovering bird populations* (O1) received the highest global priority among all SWOT factors, closely followed by the strength *Fewer collisions* (S1). Although homeowner priorities for weaknesses and threats were lower than for strengths and opportunities, all threats and some weaknesses still received higher global priorities than the strengths *Fewer people witnessing collisions* (S2) and *Fewer carcasses to clean up* (S3). The opportunity *Recovering bird populations* (O1) followed by the strength *Fewer collisions* (S1) also received the two highest global priorities for conservation practitioners. Additionally, this group prioritized weaknesses over threats while homeowners ranked these categories in the opposite order.

Although the two groups had similar broad priorities, such as valuing strengths and opportunities over weaknesses and threats, conservation practitioners gave higher priority to the top factor in some categories, suggesting stronger perceptions toward these factors. Specifically, although *Recovering bird populations* (O1) was the highest global priority among all SWOT factors for both stakeholder groups, it received a greater global priority value for conservation practitioners (0.23) than homeowners (0.14). Similarly, the top strength (and second highest global priority among all SWOT factors) for both stakeholder groups *(Fewer collisions*; S1*)* received a greater global priority value for conservation practitioners (0.15) than for homeowners (0.11) (Table 2). Global priorities also illustrated that both homeowners and conservation practitioners gave low priority to *Fewer people witnessing collisions* (S2) and *Fewer carcasses to clean up* (S3) relative to other strengths and many other weakness and threats.

## Potential for conflict and strength of opinions within stakeholder groups

Regarding potential for conflict indices ($PCI_2$) for Survey 2, comparison of the bubbles for homeowners (Fig 3A) and conservation practitioners (Fig 3B) for each pairwise comparison illustrates there was more conflict among responses for homeowners than conservation practitioners for 4 of 6 comparisons. Additionally, relative locations of bubbles on the y-axis (which indicates the difference in preference for each priority in a pairwise comparison) illustrate that homeowners were more neutral than conservation practitioners for all 6 pairwise comparisons.

## Discussion

Our results suggest that both homeowners and conservation practitioners have an overall positive perception toward potential benefits related to bird-window collision mitigation and prevention measures. This indicates stakeholders may believe that benefits of implementing management to reduce bird-window collisions outweigh any obstacles that may impede such measures. Although generally similar in their positive views, the two stakeholder groups displayed some differences in their specific priorities regarding strengths, weaknesses, opportunities, and threats surrounding this issue. Specifically, homeowners gave greater priority than conservation practitioners to threats, indicating more concern among homeowners about external obstacles (financial and policy related) that may impede bird-window collision management efforts.

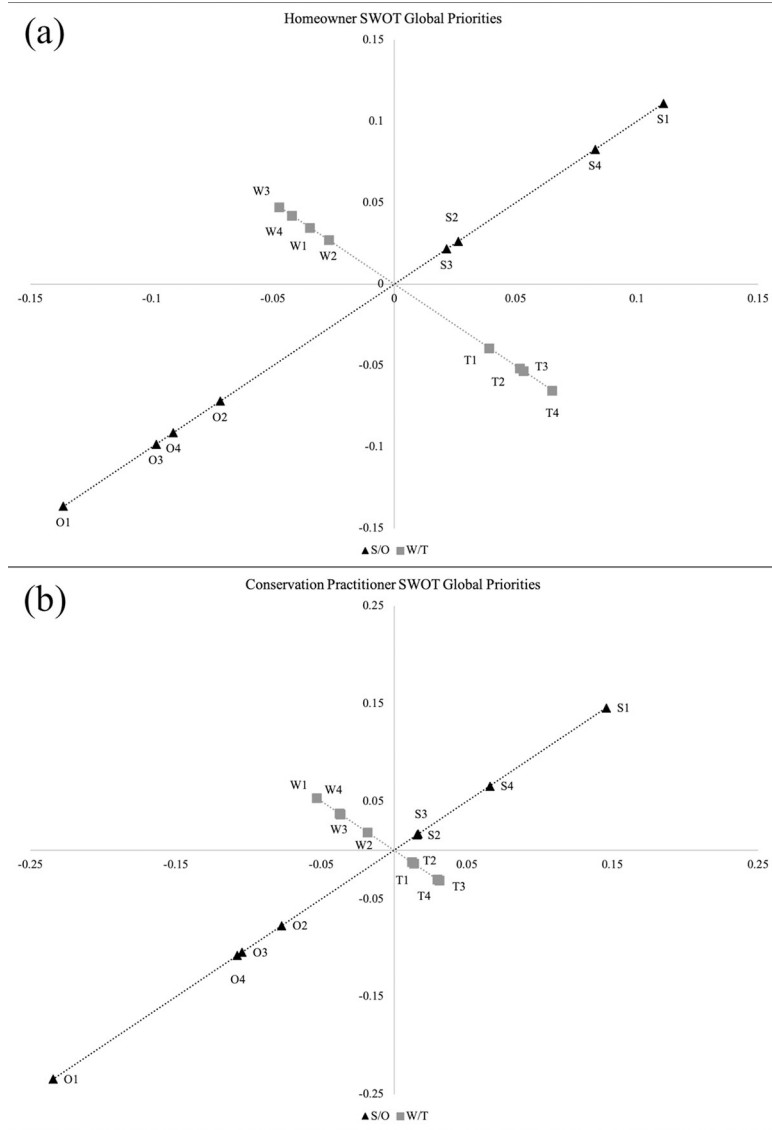

**Fig 2. Perception maps of SWOT global priorities for each stakeholder group.** Perception maps illustrating homeowner (a) and conservation practitioner (b) strength, weakness, opportunity, and threat (SWOT) global priorities for a study evaluating perceptions about potential outcomes of bird-window collision mitigation and prevention. Factors with the highest global priority are farthest from the origin. **S1**: Fewer collisions; **S2**: Fewer carcasses to clean up; **S3**: Fewer people witnessing collisions; **S4**: Fewer stunned birds that die of other causes while recovering from colliding. **W1**: No economic incentives for building for bird-friendly buildings; **W2**: Lack of architect experience in bird-friendly design; **W3**: Lack of availability of expert consultation for bird-friendly design; **W4**: Financial burden of treating glass or including bird-friendly design in building process. **O1**: Recovering bird populations; **O2**: Public exposure to bird-friendly options; **O3**: Consideration of birds in building design becoming a norm/standard; **O4**: Greater energy efficiency of buildings. **T1**: Unknown social acceptance of bird-friendly treatments and design; **T2**: Lack of understanding of federal/state policy on bird-window collisions; **T3**: Reduced resources available to spend on other facilities maintenance/improvements; **T4**: No federal/state policy in many areas.

## Stakeholder perceptions about bird-window collision management

Results indicate that the homeowner and conservation practitioner groups, while in general agreement on their positive perceptions about managing bird-window collisions, each have unique aspects of their perceptions that are important to consider in order to make headway

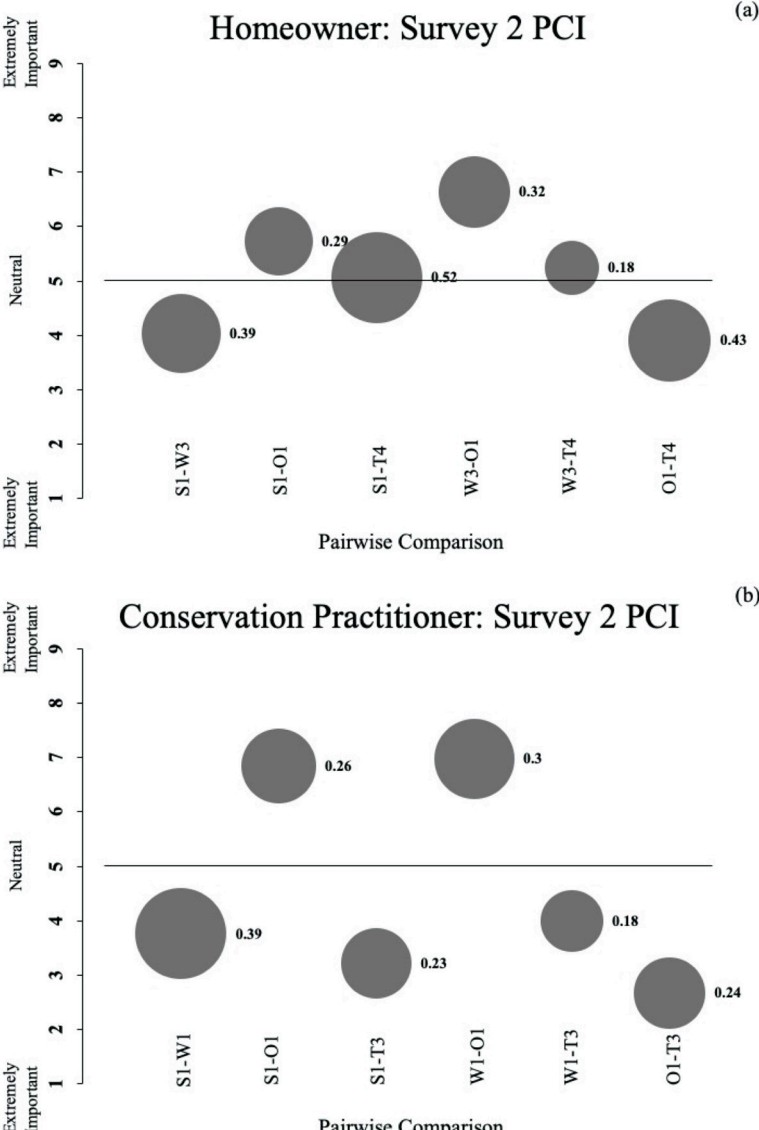

**Fig 3. Potential for conflict indices from survey 2 for each stakeholder group.** Illustration of the potential for conflict index (PCI$_2$) based on homeowner (a) and conservation practitioner (b) responses to Survey 2 in a study evaluating perceptions about potential outcomes of bird-window collision mitigation and prevention. Bubble size and values correspond and indicate the dispersion (conflict) among respondent answers (larger bubbles/numbers indicate greater conflict). The location of the bubble indicates the scale mean or the direction respondents lean in their answers to pairwise comparisons (e.g, 5 indicates completely neutral; values lower and higher than 5 indicate more non-neutral perceptions). Each bubble is an individual pairwise comparison indicated by the labels. Pairwise comparisons correspond visually to the y-axis scale (e.g., for S1-W3, 1 corresponds to S1 and 9 corresponds to W3). For a description of all strengths (S), weaknesses (W), opportunities (O), and threats (T), see Table 1.

in addressing this conservation issue. As evidenced by the PCI analysis, homeowners had more conflict in their responses to pairwise comparisons than conservation practitioners, indicating differing opinions within the group. PCI analysis also indicated that homeowners were more neutral than conservation practitioners in their responses, demonstrating differing or a potential lack of strong opinions within the group. Although we provided contextual information about this project in the survey's introductory materials, a lack of prior knowledge about

the issue—which was anecdotally revealed from comments made by gateway contacts in the homeowner group—could have contributed to their relatively neutral perceptions and conflicting responses. The less-conflicting responses within the conservation practitioner group could be due to greater knowledge about the issue or more cohesion within the group due to a shared field of profession and its associated sources of information. Specifically, those in the field of wildlife conservation likely have greater, and perhaps more consistent, exposure to major bird conservation issues through training opportunities, professional conferences, social media networks, newsletters, and scientific publications. It is important to note that the homeowner group included gateway contacts from a wide variety of professional backgrounds, which could explain the lesser degree of agreement within the group.

As evidenced by high group priority values for the strength and opportunity categories, as well as high global priority values for individual strengths and opportunities, our results indicate that both stakeholder groups have positive views about bird-window collision mitigation and prevention measures. Members of these groups may therefore be willing to participate in or support implementation of measures to reduce bird collisions. Because the top ranked strengths and opportunities capture outcomes related to bird conservation and welfare (e.g., recovering bird populations), not anthropocentric benefits (e.g., no longer having to clean up or observe collisions), our results suggest that stakeholders value mitigating and preventing collisions for the sake of the birds themselves. This result demonstrates that stakeholders may have a general sense of caring and responsibility for birds—and/or that they view birds as aesthetically, culturally, or economically valuable [56, 57]—which lends additional support to the potential acceptability and implementation of management measures. Due to a greater degree of neutrality and lack of strong opinions within the homeowner group (as illustrated by the PCI), and because some homeowners in our study were not previously aware of bird-window collisions and underlying challenges, our findings suggest a strong need for public education on this issue.

Advantageously, the positive perceptions about reducing bird-window collisions, and the apparently bird-centric reasons behind these positive perceptions, suggest that members of the public may be receptive to further education about this issue. Menacho-Odio [31] also investigated public perception and knowledge of bird-window collisions in Monteverde, Costa Rica, and concluded that while participants had general knowledge of the issue, few were aware of the magnitude of the problem. This previous study recommended targeted education that informs people about the large number of bird-window collisions that occur, as well as effective methods for preventing collisions. There are multiple publicly available resources from which individuals can learn about bird-window collisions and ways to reduce them. For example, the American Bird Conservancy (ABC) has published a website geared toward the public [58], a Bird-Friendly Building Design booklet targeting all types of building owners and managers, as well as architects [22], interactive web resources and educational materials for homeowners and architects, and a framework to help policy makers develop ordinances and legislation to reduce collisions. Similar and complementary resources to improve stakeholder knowledge about bird-window collisions have also been developed by other conservation organizations and agencies (e.g., USFWS 2021; National Audubon Society 2021; FLAP Canada 2019) [59–61]. While many resources are available, active education on this topic would also be beneficial. Specifically, increased funding and staffing to expand the delivery and interpretation of such resources to stakeholders, along with research to improve understanding of how best to develop and distribute these resources to ensure they are used, are needed to make further headway in reducing bird-window collisions.

As evident from the factor and global priority values for threats, homeowners highly prioritized policy-related obstacles to bird-window collision mitigation and prevention. However,

importantly, multiple states, cities, and municipalities across North America have already enacted policies designed to reduce bird-window collisions, including San Francisco, California, U.S.A. [24] and Minnesota, U.S.A. [25]. The U.S. House of Representatives also approved legislation (Bird Safe Buildings Act of 2021) that would require bird-friendly measures at many new and renovated U.S. federal buildings; however, this act has not yet passed the U.S. Senate [27]. Thus, while there is concern among homeowners about potential policy-related obstacles, many may not know that relevant policies already exist. This points again to the importance of education, as increasing awareness of existing and proposed policies could increase support for them among the public, and therefore, among policymakers.

Beyond educating homeowners about existing and planned policies related to bird-window collisions, homeowners could also be informed that implementing bird-friendly measures at homes might be their responsibility even with policies in existence. To date, no legislation and policies have focused on residential structures, and the proposed U.S. federal bill only focuses on public buildings. Thus, there are no formal mechanisms to ensure that collisions are reduced at residences, even though residences collectively cause a large proportion of total bird collisions [5, 7]. Although public education may encourage some homeowners to expend their own resources on measures to reduce bird-window collisions, formal programs to encourage these actions may be necessary to ensure that a large proportion of homes become bird-friendly in the future, especially for lower income residents that lack expendable resources to pay for such measures. Examples of such potential programs include conservation grants/subsidies that help pay for materials that make existing windows more bird-friendly, and revisions to existing sustainability or wildlife-friendly certification programs to specifically incorporate considerations related to reducing bird-window collisions.

Our analysis identified other potential barriers to widespread bird-window collision management. For example, homeowners identified a lack of availability of expert consultation as another top threat. Although the above-mentioned education campaigns could help empower homeowners to reduce collisions themselves, this result suggests that widespread adoption of collision management practices at homes may require increased training of consultants and outreach professionals that convey information about collision management. Conservation practitioners identified a lack of resources available to spend on other facilities/maintenance improvements as a top threat arising from the costs of collision management. In addition to emphasizing the need for low-cost management options, this result suggests that approaches that reduce collisions while meeting other facilities-related needs may be especially likely to be adopted. Notably, some approaches that are highly effective in reducing bird-window collisions, including reducing nighttime lighting [14] and some of the films, coatings, and decals adhered to windows to make them more visible [22], also may contribute to reducing building-related energy costs. Communicating the dual benefits of such approaches may lead to greater adoption of bird-friendly building management techniques.

### Limitations and future research

While this research provides valuable information to advance efforts to manage bird-window collisions, there were some limitations and potential biases related to our analyses. We were, for example, unable to analyze perspectives of architects as an independent stakeholder group due to limited recruitment for participation in our surveys. Architects are a crucial stakeholder in the issue of bird-window collisions, and further research should seek to thoroughly evaluate their perceptions about this topic. The low number of respondents for architects leads to the question of how best to reach and engage with this stakeholder group. Potential routes to engage architects include having bird-window collision researchers present at architectural

society conferences, creating publication materials geared toward architects, or reaching out directly to architectural societies or firms about bird-window collisions.

Another limitation concerns the representativeness of our sample of survey respondents, which relates both to the limited sample size of respondents and mixed-data collection approach that used gateway contacts and recruitment through social media platforms. Notably, the AHP approach does not require large sample sizes to result in statistically robust results that are useful for understanding stakeholder perceptions and informing management decisions [62]. Instead, reliability of results from this approach is interpreted using consistency ratios, which indicate the degree of consistency of responses within stakeholder groups. Consistency ratios for groups used in our analyses were considered acceptable [63], suggesting our results are reliable. However, because many of the gateway contacts we recruited for the homeowner group were our personal and professional contacts, our sample of homeowners could have been biased toward bird enthusiasts rather than providing full representation of the diversity within this group. Nonetheless, our homeowner sample contained many respondents beyond the gateway contacts that we did not know personally, indicating that there may have been variation in levels of interest or support for bird-window collision management and wildlife conservation more broadly. Although our approach does not require large sample sizes, we caution against making broad generalizations from our results, especially for the homeowner group, due to these potential issues regarding sample representativeness.

Our results lay a foundation for future research into stakeholder perceptions, priorities, and potential disputes and conflicts related to bird-window collision management. Conducting research to better understand motivations and barriers to behavioral change will be crucial for designing collision management programs that garner broad support and participation from the public. In this study, we examined stakeholder perceptions and priorities, but other important factors that influence behavioral changes (e.g., social and cultural norms, institutional and economic factors) should also be evaluated [64]. Further, research that identifies social-psychological barriers that may lead to conflicts among groups (e.g., conservation organizations recommending collision management approaches vs. building management entities resistant to recommendations) could facilitate more-rapid adoption of bird-friendly building design, and similar research related to the green building movement may be instructive for this issue [65]. We did not collect demographic information from respondents, nor did we know the geographic representation of our sample other than for gateway contacts. Because the factors that influence behaviors, perceptions, and conflicts can vary regionally and among demographic groups (e.g., among different age groups), future research could evaluate how perceptions about bird-window collisions vary regionally and in relation to various demographic factors.

Another essential area of future research is to evaluate stakeholders' willingness to pay (WTP) for measures to reduce collisions. Our study shows that the stakeholder groups we evaluated are receptive to bird-window collision management, but that does not necessarily translate into a willingness to pay for these measures, especially if doing so at private residences is the responsibility of homeowners. Past research evaluating WTP for conservation practices indicates that the public is often receptive to wildlife conservation and willing to pay for it [66–69]. The public's WTP for conservation practices can be heavily influenced by sense of place, or the value and meaning that individuals attach to a physical location [70, 71]. This suggests that informational materials that tie the issue of bird-window collisions to an individual's location or experience may be a particularly effective way to increase WTP. For example, educational materials could highlight the likely number of collisions that occur in areas where residents live and how collisions may be affecting locally important bird species. Another study found that while members of the public were willing to pay for bird conservation, they

believed the government should also play a role [68], a finding that lends additional support to grant, subsidy, and/or certification programs specifically geared toward reducing bird-window collisions. Although homeowners are a critical group to examine with regard to WTP to reduce bird-window collisions, other stakeholders such as business owners and agencies operating in larger buildings are also important stakeholders to study.

Birds face multiple human-related threats, including climate change, habitat loss, and other direct mortality sources (e.g., cat predation, other types of collisions) [3]. While it is important to investigate bird-window collisions specifically, understanding human perceptions of other threats is also necessary because this may lead to insights about which conservation actions are most and least likely to be supported and implemented by the public. Understanding perceptions of different threats, as well as willingness to pay and/or willingness to change behaviors in ways that mitigate these threats, could also lead to more effective conservation strategies that optimize the tradeoff between addressing the most substantial threats and addressing the threats for which substantial management inroads are possible.

## Conclusions

This study provides novel insight about how important stakeholder groups view and prioritize benefits and obstacles related to bird-window collision mitigation and prevention. Our research suggests that substantial advances can be made to reduce bird-window collisions because both homeowners and conservation practitioners had positive views, suggesting their receptivity toward and acceptability of collision management measures. However, because of the more neutral views and more conflicting responses within the homeowner group, our results also highlight the importance of targeting these stakeholders with education materials that provide information about bird-window collisions and policies and publicly available solutions that reduce them. Homeowners are a critical stakeholder group because a large proportion of collisions occur at residential buildings; having their support and participation in bird-window collision mitigation and prevention could help significantly reduce collisions. Future research needs related to human dimensions of bird-window collisions and other avian mortality sources include evaluating perceptions of other stakeholder groups (e.g., architects and policymakers), studying social-psychological barriers to reducing collisions, determining willingness to pay for collision mitigation and prevention, and clarifying relative perceptions about impacts and management of human-related threats other than bird-window collisions. Because bird-window collisions are a human-caused phenomenon, understanding human perspectives and priorities about this issue will be crucial to addressing this threat and thus benefitting bird populations.

## Supporting information

**S1 File. SWOT Survey 1.** Strengths, weaknesses, opportunities, and threats (SWOT) survey distributed to all respondents (i.e., Survey 1 described in main text) consisting of all pairwise comparisons between factors in each SWOT category using a scale of one to nine. For this survey, all possible pairwise comparisons were made between factors within (but not between) each SWOT category (e.g., all strengths compared to each other, but strengths not compared to weaknesses, opportunities, and threats). Analysis of responses to this survey revealed top-ranked SWOT factors in each category, which were unique to each stakeholder group and used to generate comparisons in Survey 2.
(PDF)

**S2 File. SWOT Survey 2 for homeowners.** Strengths, weaknesses, opportunities, and threats (SWOT) survey distributed to respondents in the homeowner stakeholder group (i.e., Survey 2 for homeowners described in main text) based on their responses to Survey 1. For this survey, all possible pairwise comparisons were made between the top-ranking factors from each SWOT category for homeowners (e.g., top homeowner strength compared to top weakness, opportunity, and threat).
(PDF)

**S3 File. SWOT Survey 2 for conservation practitioners.** Strengths, weaknesses, opportunities, and threats (SWOT) survey distributed to respondents in the conservation practitioner stakeholder group (i.e., Survey 2 for conservation practitioners described in main text) based on their responses to Survey 1. For this survey, all possible pairwise comparisons made between the top-ranking factors from each SWOT category for conservation practitioners (e.g., top conservation practitioner strength compared to top weakness, opportunity, and threat).
(PDF)

**S1 Dataset. SWOT and PCI data analysis.** This file contains all response data generated from strengths, weaknesses, opportunities, and threats (SWOT) Surveys 1 and 2 (see main text and S1–S3 Files for details about these surveys) along with data and analysis for the potential for conflict index (PCI).
(XLSX)

# Acknowledgments

We thank Christine Sheppard, Daniel Klem, and Stephen Hager for providing preliminary rankings of candidate SWOT factors, Samantha Cady, Jared Elmore, and Timothy O'Connell for insightful feedback on methods and an earlier version of the manuscript, and all survey respondents for their participation. We also thank the handling editor and two anonymous reviewers for their constructive feedback and suggestions that greatly improved the manuscript.

# Author Contributions

**Conceptualization:** Georgia J. Riggs, Omkar Joshi, Scott R. Loss.

**Data curation:** Georgia J. Riggs.

**Formal analysis:** Georgia J. Riggs.

**Funding acquisition:** Scott R. Loss.

**Investigation:** Georgia J. Riggs, Scott R. Loss.

**Methodology:** Georgia J. Riggs, Omkar Joshi.

**Supervision:** Omkar Joshi, Scott R. Loss.

**Validation:** Omkar Joshi.

**Writing – original draft:** Georgia J. Riggs.

**Writing – review & editing:** Omkar Joshi, Scott R. Loss.

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
