## [Decision Letter · Decision Letter 0]

10 Aug 2021

PONE-D-21-17993

Stakeholder perceptions of bird-window collisions

PLOS ONE

Dear Dr. Riggs,

Thank you for submitting your manuscript to PLOS ONE. After careful consideration, we feel that it has merit but does not fully meet PLOS ONE’s publication criteria as it currently stands. Therefore, we invite you to submit a revised version of the manuscript that addresses the points raised during the review process. Specifically, both reviewers found merit in the work, but had a variety of items that need attention as noted in their comments below.

We look forward to receiving your revised manuscript.

Kind regards,

Christopher A. Lepczyk

Academic Editor

PLOS ONE

Reviewers' comments:

Reviewer's Responses to Questions

**Comments to the Author**

1. Is the manuscript technically sound, and do the data support the conclusions?

Reviewer #1: Yes

Reviewer #2: Partly

2. Has the statistical analysis been performed appropriately and rigorously? 

Reviewer #1: Yes

Reviewer #2: Yes

3. Have the authors made all data underlying the findings in their manuscript fully available?

Reviewer #1: No

Reviewer #2: No

4. Is the manuscript presented in an intelligible fashion and written in standard English?

Reviewer #1: Yes

Reviewer #2: Yes

5. Review Comments to the Author

Reviewer #1: This manuscript investigates an understudied topic: the human dimensions of reducing bird-window collisions. In general, I think the manuscript is well-written, however I have several comments about areas where more information is needed.

Abstract

Line 36: Note that “implement” should be “implementing”

Link 37-40: Because you didn’t analyze the effectiveness of an education program, I don’t think your results actually show this: “that targeted and active education may be successful in garnering public support for and participation in bird-window collision mitigation and prevention.” I suggest instead focusing your final sentence on how while stakeholders focused on the benefits/opportunities, there were some obstacles that were identified that could potentially be addressed moving forward.

Introduction

Line 71: Might be useful mentioning in this paragraph the fact that people have to adopt new technologies in their homes to prevent bird collisions, which is another critical argument for why its important to look at people’s perceived barriers/opportunities

Line 79: I don’t think it needs to be mentioned that it was a graduate thesis study.

Line 82-84: To me, this sentence is awkward as it basically saying that learning about the public’s perception is important because it tells us about public perception. Suggest revising to be more specific about why understanding public perspectives and acceptance is important (I realize you get into this in the next sentence).

Methods

Link 105-106: Can you give a few examples here of how it has been used?

Line 114: I think a more in-depth description of what AHP is and how it has been used would be helpful to the reader here.

Line 135: Suggest editing to “we used snowball sampling, a nonprobability sampling method…” same with description of purposive sampling below.

Line 135: How many gateways respondents in each category did you start off with in each group for snowballing?

Line 153: “we decided to use this approach prior to collecting any data”- this confuses me because purposive sampling is a type of data collection

Line 160: Can you explain here why there were two surveys and did all stakeholders take both surveys?

Line 167-168: How did you develop this list, and how did you make sure it was comprehensive? Also I think reiterating what is meant by strengths/weaknesses here is helpful- it’s hard for me to immediately understand how “strength” or “weaknesses” applies to analyzing public perceptions on a conservation issue- language such “benefits of addressing the issue” or “challenges to addressing the issue” is much more intuitive. I also think clarifying here the difference between strengths and opportunities and weaknesses and threats would be helpful.

Line 178-179: So just to clarify: all participants ranked each SWOT factor in each category against one another?

A full copy of your survey questions would be helpful.

Lines 182-184: This seems like an undetailed summary of your data analysis section. I’d suggest deleting it as it only raises more questions for the reader before the reader gets to the data analysis section.

Line 193: Why was each survey tailored to each group? Is this standard practice? Why would you have stakeholders compare between SWOT categories? What does that tell you theoretically or practically? Is this the AHP portion of the framework? Reading this methods section makes me realize that some very clear research questions that match specifically with survey design at the end of the introduction would be very helpful to a reader to understand why you are doing what you are doing in the methods/survey design.

Data Analysis

Line 214: How did you determine the mean from a pairwise comparison? This is where seeing your exact survey questions would be useful. Did you code as yes/no selected in the pairwise comparison and the mean is then of the binary variable? Some more handholding for the reader on this analysis would be helpful.

Line 229-231: I’m somewhat confused here- if consistency was high between architect and NGO practitioner groups, why did you drop architects and not NGOs? And why did you then combine government and NGOs? What were the sample sizes here?

Line 240: Is PCI usually used with this sort of SWOT analysis and does it work with the way the questions were asked in the survey? I thought it’s mainly used to analyze differences in responses to direct attitude/belief metrics, but I’m no expert in PCI. Nevertheless, some more info on this in the text would be beneficial.

Results

Lines 269-261: Not sure what is meant by this: “this may be an underestimate of the geographical scope of our study as the snowball sampling approach extended to people beyond our immediate circle.” Didn’t you ask in your survey where people are from so shouldn’t you be able to provide an exact estimate of the geographical scope?

Need to include final number of survey respondents per category and per survey in the results. Also basic demographic information for the survey respondents would be helpful.

Line 275- 278: awkwardly phrased

I think a row name is needed for the total global priority values in Table 2.

Discussion

-Line 466-468: just because the sample contained many respondents you didn’t know personally doesn’t mean the sample was representative or varied compared to the population. Is there any data you can provide on how representative your sample of homeowners in particular is?

-Some more on the barriers/challenges identified and how they can be overcome would be useful in the discussion

-Future work could also look at the social-psychological barriers and what stakeholders identify as the barriers to implementing actions to address this threat.

Reviewer #2: This is an interesting and well written manuscript that analyzes stakeholder (homeowners and conservation practitioners) perceptions of bird-window collisions and possible mitigation activities.

My key concerns about this manuscript revolve around the survey sampling utilized. While the authors note that their homeowner sample may be "self-selected" in that people more interested in birds or related issues were likely more apt to respond, another key issue is that their sampling is based entirely on non-probability selection. The authors attempt to make comments reflecting perceptions of the "general public" and even "conservation practitioners" as a whole, but they really have little basis to do this. They simply do not know who their sample represents because they did not draw the sample based on probability sampling. This is a substantial limitation that, at a minimum, needs to be highlighted and discussed. Similarly, there is no way to determine, of all those who received the invitation to do the survey, what proportion of them went ahead and completed the survey. What if only 1% of those who received the invitation went ahead to do the survey? This would not even be representative of those who got the invitation, much less of any broader "general public," "homeowner" or "conservation practitioner" grouping.

Relatedly, while it is not specifically stated, can it be assumed that Survey 2 was sent to ALL those who responded to Survey 1? If so, what was the overall response? What proportion of those who completed Survey 1 also completed Survey 2? If this is a small proportion, then this is also a substantial limitation. (In fact, why are there no sample sizes reported in the manuscript??)

6. PLOS authors have the option to publish the peer review history of their article (what does this mean?). If published, this will include your full peer review and any attached files.

Reviewer #1: No

Reviewer #2: No

---

## [Author Response · Author response to Decision Letter 0]

25 Oct 2021

Reviewer #1: This manuscript investigates an understudied topic: the human dimensions of reducing bird-window collisions. In general, I think the manuscript is well-written, however I have several comments about areas where more information is needed.

Abstract

Line 36: Note that “implement” should be “implementing”

RESPONSE: We made the suggested change.

Link 37-40: Because you didn’t analyze the effectiveness of an education program, I don’t think your results actually show this: “that targeted and active education may be successful in garnering public support for and participation in bird-window collision mitigation and prevention.” I suggest instead focusing your final sentence on how while stakeholders focused on the benefits/opportunities, there were some obstacles that were identified that could potentially be addressed moving forward.

RESPONSE: We revised this final section of the abstract following the reviewer’s recommendation. First, we revised the existing sentence to avoid implying that our results evaluated effectiveness of education programs related to this issue. Second, we added an additional sentence highlighting that further efforts may be needed to overcome potential political and financial hurdles to widespread collision management. The revised sentences are at Lines 42-45 in the revised manuscript.

Introduction

Line 71: Might be useful mentioning in this paragraph the fact that people have to adopt new technologies in their homes to prevent bird collisions, which is another critical argument for why its important to look at people’s perceived barriers/opportunities

RESPONSE: We revised to mention this additional important facet in the new sentence on Lines 87-91 (this revised sentence also addresses the below comment regarding Lines 82-84).

Line 79: I don’t think it needs to be mentioned that it was a graduate thesis study.

RESPONSE: We revised to remove reference to this being a thesis study.

Line 82-84: To me, this sentence is awkward as it basically saying that learning about the public’s perception is important because it tells us about public perception. Suggest revising to be more specific about why understanding public perspectives and acceptance is important (I realize you get into this in the next sentence).

RESPONSE: We agree and fixed this issue by revising and merging this sentence with the following sentence. We also added a phrase regarding the adoption of new technologies in buildings following the reviewer’s above comment for Line 71. The newly revised section (Lines 87-91) is:

“Clarifying how people perceive bird-window collisions, and how much they support mitigation and prevention techniques, is crucial for bird conservation because implementing effective practices generally entails adoption of new products and technologies on buildings, and therefore, requires buy-in from multiple stakeholder groups (e.g., residential homeowners, owners/managers of commercial buildings, building architects, policymakers).”

Methods

Link 105-106: Can you give a few examples here of how it has been used?

RESPONSE: Thank you for this suggestion. We revised (Lines 112-116) to clarify that the method has been used to understand and compare stakeholder perceptions concerning several issues in conservation and natural resource management, such as renewable energy, ecotourism, and land management and policy. 

Line 114: I think a more in-depth description of what AHP is and how it has been used would be helpful to the reader here.

RESPONSE: In addition to providing examples of how the merged SWOT-AHP approach has been used (see previous comment), we revised (Lines 126-133) to add greater detail about AHP. Specifically, as a multi-criteria decision-making tool, AHP assigns relative weights to factors of interest based on 2-way comparisons between factors (Satty 1977); this allows objective evaluation of the degree of agreement (or disagreement) between factors and generates a matrix that allows quantification of factor rankings for each stakeholder group. 

Line 135: Suggest editing to “we used snowball sampling, a nonprobability sampling method…” same with description of purposive sampling below.

RESPONSE: We revised according to this suggestion.

Line 135: How many gateways respondents in each category did you start off with in each group for snowballing?

RESPONSE: We added information about numbers of gateway contacts at Lines 154-157. 

Line 153: “we decided to use this approach prior to collecting any data”- this confuses me because purposive sampling is a type of data collection

RESPONSE: The wording was meant to reflect that this decision was made prior to collecting data. However, since it should be obvious that all of our study design decisions were made before data collection, we revised to remove this phrase and instead state: “We used this approach because…”

Line 160: Can you explain here why there were two surveys and did all stakeholders take both surveys?

RESPONSE: Thank you for the opportunity to clarify this. We revised the methods (Lines 192-210) to better clarify how data collection involved two surveys administered successively, the typical approach under the SWOT-AHP framework. For the first survey, respondents conducted pairwise comparisons between all factors within a single SWOT category. For example, respondents compared all strength factors to each other, all weakness factors to each other, etc., but they did not compare any strength factors to weakness factors, any weakness factors to opportunity factors, etc. For the second survey, the highest ranked factors for each category (calculated using Survey 1 responses) were compared; for example, respondents compared the top strength to the top weakness, the top weakness to the top threat, etc. Notably, for Survey 2, we contacted all participants who received the Survey 1 invitation (this approach is allowed under SWOT-AHP best practices); however, we requested that they only complete Survey 2 if they had completed Survey 1. In addition to greatly revising our description of this two-tiered survey approach, we also added other requested clarifying information about the survey methodology (see responses to below comments), and to further clarify the methods, we moved information about the sampling dates during which the two surveys were administered to Lines 234-235. To clarify terminology, which we realized was quite confusing at times due to inconsistent use of terms, we also revised to more consistently refer to SWOT categories (the four major groups, including strengths, weaknesses, opportunities, and threats) and SWOT factors (the specific items within each SWOT category). Additional information about survey participation, completion, and anonymity are also now included at Lines 223-231. Finally, we now provide the entire contents of both Survey 1 and Survey 2 (for homeowners and conservation practitioners) in the Supplemental Information (S1, S2, and S3 Appendices, respectively).

Line 167-168: How did you develop this list, and how did you make sure it was comprehensive? Also I think reiterating what is meant by strengths/weaknesses here is helpful- it’s hard for me to immediately understand how “strength” or “weaknesses” applies to analyzing public perceptions on a conservation issue- language such “benefits of addressing the issue” or “challenges to addressing the issue” is much more intuitive. I also think clarifying here the difference between strengths and opportunities and weaknesses and threats would be helpful.

RESPONSE: We agree that the terms “strength,” “weakness,” “opportunity”, and “threat” are not necessarily intuitive when applied to public perceptions; however, we used these terms to stay consistent with related SWOT-AHP literature and recommended protocols for using these approaches. We have revised to make clearer (Lines 116-120) that strengths and weaknesses capture direct positive and negative outcomes, respectively, and are therefore considered internal to the issue being investigated. In contrast, opportunities and threats respectively capture positive and negative indirect outcomes external to the issue. Given our clearer definition of these terms at the beginning of the methods, as well as our examples of specific strengths, weaknesses, opportunities, and threats related to bird-window collision management (see text at Lines 197-210 and Table 1 which lists all SWOT factors that survey respondents were asked to compare), we believe we have addressed the reviewer’s concerns about making these terms as intuitive as possible. Further, to clarify terminology, which we realized was quite confusing at times due to inconsistent use of terms, we also revised to more consistently refer to SWOT categories (the four major groups, including strengths, weaknesses, opportunities, and threats) and SWOT factors (the specific items within each SWOT category).

Related to the comment about developing our final list of strengths, weaknesses, opportunities, and threats, we clearly defined these factors (and gave examples) to the group of subject-matter experts who had extensive knowledge about bird-window collisions and were asked to create a list of SWOT factors associated with this issue. Although there is no way to confirm whether this list is entirely exhaustive, we believe that our approach of seeking input from multiple experts should have identified most if not all of the major direct and indirect implications of collision management. We then ranked these lists to determine the final list of SWOT factors included in our Surveys. We note that this description of our approach to develop the final SWOT list is included in the Methods at Lines 178-185. 

Line 178-179: So just to clarify: all participants ranked each SWOT factor in each category against one another?

RESPONSE: Your interpretation is correct. As described above, we have revised to further clarify this detail at the end of that sentence (Lines 197-202).

A full copy of your survey questions would be helpful.

RESPONSE: We have added the full contents of Survey 1 and Survey 2 for both homeowners and conservation practitioners in the Supplemental Information (S1, S2, and S3 Appendices, respectively).

Lines 182-184: This seems like an undetailed summary of your data analysis section. I’d suggest deleting it as it only raises more questions for the reader before the reader gets to the data analysis section.

RESPONSE: Following the reviewer’s advice, we have now provided this information in the Data Analysis section (Lines 241-253) and deleted here. 

Line 193: Why was each survey tailored to each group? Is this standard practice? Why would you have stakeholders compare between SWOT categories? What does that tell you theoretically or practically? Is this the AHP portion of the framework? Reading this methods section makes me realize that some very clear research questions that match specifically with survey design at the end of the introduction would be very helpful to a reader to understand why you are doing what you are doing in the methods/survey design.

RESPONSE: As described at Lines 202-205, please note that only Survey 2 was tailored to each stakeholder group, and this was done using the response data for each group from Survey 1. For example, calculations for homeowner and conservation practitioner responses to Survey 1 revealed different top-ranked weaknesses and threats for each group; therefore, the specific pairwise comparison of the top-ranked weakness and threat in Survey 2 was different for these two stakeholder groups. This approach is standard practice for the SWOT-AHP analytical approach and is required to obtain a global priority value for each factor, which allows overarching conclusions about whether strengths, weaknesses, opportunities, and/or threats play the biggest role in the issue of bird-window collision management. 

The reviewer’s general point about how these analyses do not necessarily clearly tie to the specific objectives stated in the Introduction is well-taken. To address this issue, we revised the wording of our objectives in both the Abstract and Introduction to make them more specific and to better match the results generated using our survey and analysis approach. Furthermore, in the Results, we also revised our section subheadings to more intuitively describe the types of conclusions facilitated by each set of calculations/analyses.

Data Analysis

Line 214: How did you determine the mean from a pairwise comparison? This is where seeing your exact survey questions would be useful. Did you code as yes/no selected in the pairwise comparison and the mean is then of the binary variable? Some more handholding for the reader on this analysis would be helpful.

RESPONSE: Thank you for identifying this lack of clarity in the manuscript. Following the SWOT-AHP protocol (Dwivedi and Alavalapati 2009, Starr et al. 2019), a geometric mean of individual responses was calculated for each pairwise comparison. As we note in Lines 192-197, the data were collected using Saaty’s 9 point Likert Scale, where respondents completed each pairwise comparison by selecting a point on the scale that corresponded to a value of 1 through 9, with a value of 1 being “extremely important” toward one factor, 9 being “extremely important” toward the other factor, and 5 being “equally important” between the two factors. We added text in Lines 244-250 to clarify how we used pairwise comparisons to calculate weighted geometric means. Further, to fully clarify the survey format including the above Likert scale, we added the entire contents of Survey 1 and Survey 2 for both homeowners and conservation practitioners in the Supplemental Information (S1, S2, and S3 Appendices, respectively).

Line 229-231: I’m somewhat confused here- if consistency was high between architect and NGO practitioner groups, why did you drop architects and not NGOs? And why did you then combine government and NGOs? What were the sample sizes here?

RESPONSE: Please note that here we are referring to measures of response consistency within stakeholder groups, which we have revised to better clarify at Lines 257-260). Both the NGO practitioner and government practitioner groups had unacceptably low within-group consistency ratios. Rather than excluding data from both groups, and because these groups are relatively similar (i.e., respondents from both generally have training and backgrounds in ecology, conservation biology, and/or natural resources management), we merged them. For the architect group, which also had an unacceptably low within-group consistency ratio, there were no other similar groups so we had to exclude data from this group. Related to the sample size comment, we also revised this section to add sample sizes for each stakeholder group. 

Line 240: Is PCI usually used with this sort of SWOT analysis and does it work with the way the questions were asked in the survey? I thought it’s mainly used to analyze differences in responses to direct attitude/belief metrics, but I’m no expert in PCI. Nevertheless, some more info on this in the text would be beneficial.

RESPONSE: The reviewer is correct that PCI is not often used with SWOT-AHP. However, as we have revised to make clearer in the Methods (Lines 284-288), the goal of using PCI was to provide additional insights related to potential conflicts of opinion within stakeholder groups, and related to collective strengths (vs. neutrality) of group opinions. To help better frame and introduce the PCI analysis, which admittedly seemed somewhat ad hoc in the previous version of the manuscript, we added a new 3rd objective to the Introduction (Lines 98-99) describing our goal of understanding within-group conflicts and strength of opinions. Despite not being frequently used with SWOT-AHP, there is no reason from a statistical point of view that PCI cannot be applied to this type of survey, since the survey questions are bipolar scalar with a neutral value. 

Results

Lines 269-261: Not sure what is meant by this: “this may be an underestimate of the geographical scope of our study as the snowball sampling approach extended to people beyond our immediate circle.” Didn’t you ask in your survey where people are from so shouldn’t you be able to provide an exact estimate of the geographical scope?

RESPONSE: Because we recruited our gateway contacts directly, we knew their geographic locations (these contacts represented at least 18 U.S. states, as well as Canada), and as is typical of the snowball sampling protocol, we asked our gateway contacts to assist us in identifying other respondents. However, we were unable to collect location information from those other, additional respondents because our surveys were anonymized to minimize the collection of personally identifiable information. We revised this section of the Results (Lines 305-309) to better clarify the geographic distribution of gateway contacts, and the reason that we cannot provide the exact geographical distribution of all respondents. 

Need to include final number of survey respondents per category and per survey in the results. Also basic demographic information for the survey respondents would be helpful.

RESPONSE: We agree with the need to add sample size information, but instead of adding it in the Results, we revised to include sample sizes in the Data Analysis section of the Methods, where the topic of group sample sizes is first mentioned (see Lines 260-268 in the Methods). However, consistent with the previous SWOT-AHP literature, we did not collect demographic information from survey respondents, as it was not directly relevant to the goals of our study. 

Line 275- 278: awkwardly phrased

RESPONSE: We fully revised the caption for Table 2: “Summary of factors used in strengths, weaknesses, opportunities, and threats (SWOT) analyses related to perceptions and potential outcomes of bird-window collision mitigation and prevention. Factor priority, global priority, and group priority values (see text for description of these values) are illustrated for two different stakeholder groups.”

I think a row name is needed for the total global priority values in Table 2.

RESPONSE: We added row descriptions to identify group priority values (i.e., summed global priority values across all strengths, weaknesses, opportunities, and threats for each stakeholder group).

Discussion

-Line 466-468: just because the sample contained many respondents you didn’t know personally doesn’t mean the sample was representative or varied compared to the population. Is there any data you can provide on how representative your sample of homeowners in particular is?

RESPONSE: Thanks for the opportunity to clarify related to the issue of survey representativeness. We have added new text to more fully discuss the representativeness of our sample and all of the related issues we highlight below in this response (see Lines 538-552). The reviewer is correct that we could not ensure representativeness; however, our work is grounded on the theoretical premises of the AHP approach, which does not require large or representative samples to achieve statistically robust results that are useful for understanding stakeholder perceptions and informing decision-making (e.g. Darko et al, 2017). Many examples exist of using the AHP protocol to generate conclusions about natural resource-related issues based on a small number of respondents. For example, Shrestha et al. (2004) used opinions of three respondents to explore the potential for adoption of silvopasture methods in Florida, Kurtitila et al. (2000) used opinions of two respondents to compare strengths, weaknesses, opportunities, and threats associated with forest certification in Finland, and Masozera et al. (2006) used 11 total participants from three groups (5 from one group and 3 in each of the other two groups) in their assessment of community forestry in Rwanda. Many other similar examples also exist (e.g. Margles et al. 2010, Dwivedi and Alavalapati 2009). 

Further, it is important to note that the consistency ratio, not the sample size or sample size distribution, is the relevant metric for determining the reliability of AHP results (e.g., Shrestha et al. 2004). In our study, for all except 2 SWOT categories in Survey 1, consistency ratios were <10%, indicating consistent responses within stakeholder groups. For conservation practitioners, the weaknesses and opportunities SWOT categories had consistency ratios of 19% and 18%, respectively, indicating some inconsistency. While consistency values less than 10% are considered excellent, consistency ratios >20% require re-examination (Margles et al, 2010). Since most of our consistency ratios were less than 10% and all of them were less than 20%, re-examination was not warranted and our results should be reliable.

Another point regarding our sampling framework is worth highlighting here. Although we recruited gateway contacts who may have been knowledgeable about the subject matter, the additional respondents recruited via snowball sampling, as well as the homeowners we recruited through purposive sampling, may not have been fully aware of the bird-window collision issue. Therefore, to make sure all respondents were fully aware of this issue, we provided introductory information about bird-window collisions at the beginning of Survey 1 (see the newly added S1 Appendix for the exact text provided, as well as the new text in the Methods at Lines 221-223 clarifying that this information was provided in the introductory materials of the survey).

Some more on the barriers/challenges identified and how they can be overcome would be useful in the discussion

RESPONSE: We added an additional paragraph to the Discussion (see Lines 511-525) covering additional top barriers/challenges identified by the homeowner and conservation professional groups (specifically, a lack of available expert consultants for the homeowner group and a lack of funding to spend on other facilities/building-related needs for the conservation practitioner group), as well as potential approaches to overcome these challenges. 

-Future work could also look at the social-psychological barriers and what stakeholders identify as the barriers to implementing actions to address this threat.

RESPONSE: We added a sentence in the Research Needs paragraph (Lines 553-559) recommending research to understand social-psychological barriers, and we highlight that such research for the similarly-focused green building movement may be instructive for this issue.

Reviewer #2: This is an interesting and well written manuscript that analyzes stakeholder (homeowners and conservation practitioners) perceptions of bird-window collisions and possible mitigation activities.

My key concerns about this manuscript revolve around the survey sampling utilized. While the authors note that their homeowner sample may be "self-selected" in that people more interested in birds or related issues were likely more apt to respond, another key issue is that their sampling is based entirely on non-probability selection. The authors attempt to make comments reflecting perceptions of the "general public" and even "conservation practitioners" as a whole, but they really have little basis to do this. They simply do not know who their sample represents because they did not draw the sample based on probability sampling. This is a substantial limitation that, at a minimum, needs to be highlighted and discussed

RESPONSE: Thank you for the opportunity to clarify regarding the related issues of sample size and representativeness. These issues were raised by both reviewers and thus warrant a thorough response. First, we note that we have revised the methods to clearly present the sample sizes of survey respondents obtained for each stakeholder group (Lines 260-268).

Second, we added new text in the Discussion (see Lines 538-552) to more fully discuss the representativeness of our sample, and all of the related issues we highlight below in this response. The reviewer is correct that we could not ensure representativeness; however, our work is grounded on the theoretical premises of the AHP approach, which does not require large or representative samples to achieve statistically robust results that are useful for understanding stakeholder perceptions and informing decision-making (e.g. Darko et al, 2017). Many examples exist of using the AHP protocol to generate conclusions about natural resource-related issues based on a small number of respondents. For example, Shrestha et al. (2004) used opinions of three respondents to explore the potential for adoption of silvopasture methods in Florida, Kurtitila et al. (2000) used opinions of two respondents to compare strengths, weaknesses, opportunities, and threats associated with forest certification in Finland, and Masozera et al. (2006) used 11 participants from three groups (5 from one group and 3 in the other two groups) in their assessment of community forestry in Rwanda. Many other similar examples also exist (e.g. Margles et al. 2010, Dwivedi and Alavalapati 2009). 

Further, it is important to note that the consistency ratio, not the sample size or sample size distribution, is the relevant metric for determining the reliability of AHP results (e.g. Shrestha et al. 2004). In our study, for all except 2 SWOT categories in Survey 1, consistency ratios were <10%, indicating consistent responses within stakeholder groups. For conservation practitioners, the weaknesses and opportunities SWOT categories had consistency ratios of 19% and 18%, respectively, indicating some inconsistency. While consistency values less than 10% are considered excellent, consistency ratios >20% require re-examination (Margles et al, 2010). Since most of our consistency ratios were less than 10% and all of them were less than 20%, re-examination was not warranted and our results should be reliable.

Another point regarding our sampling framework is worth highlighting here. Although we recruited gateway contacts who may have been knowledgeable about the subject matter, the additional respondents recruited via snowball sampling, as well as the homeowners we recruited through purposive sampling, may not have been fully aware of the bird-window collision issue. Therefore, to make sure all respondents were fully aware of this issue, we provided introductory information about bird-window collisions at the beginning of Survey 1 (see the newly added S1 Appendix for the exact text provided, as well as the new text in the Methods at Lines 221-223 clarifying that this information was provided in the introductory materials of the survey).

Similarly, there is no way to determine, of all those who received the invitation to do the survey, what proportion of them went ahead and completed the survey. What if only 1% of those who received the invitation went ahead to do the survey? This would not even be representative of those who got the invitation, much less of any broader "general public," "homeowner" or "conservation practitioner" grouping.

RESPONSE: Good point. Consistent with the reviewer comment, we were aware that response rate would be difficult to gauge with the snowball sampling method that we used. Nonetheless, since our data analysis was based on the AHP approach, non-response bias was not a significant concern (see also our more detailed response to the previous comment). In a typical survey study, researchers seek to quantify non-response bias using methods like non-response phone calls, statistical comparisons between early and late respondents, and census comparisons. However, with AHP-based analysis, the consistency ratio is the metric used to gauge sample reliability. As highlighted in detail under the previous comment, we conducted a thorough CR analysis, the results of which suggest our results are reliable. Although we argue that our results are reliable, and we have added more nuanced discussion related to the representativeness of our survey results, we realize there may be lingering concerns related to survey representativeness. Therefore, in addition to the above text revisions, we have also included a statement cautioning readers from making broad generalizations based on our data (Lines 550-552). 

Relatedly, while it is not specifically stated, can it be assumed that Survey 2 was sent to ALL those who responded to Survey 1? If so, what was the overall response? What proportion of those who completed Survey 1 also completed Survey 2? If this is a small proportion, then this is also a substantial limitation. (In fact, why are there no sample sizes reported in the manuscript??)

RESPONSE: We have revised to add sample sizes (Line 260-268) for each stakeholder group and for both Surveys 1 and 2, but as described above, we can’t calculate response rate due to our snowball sampling approach and because we had to limit collection of personally identifiable information from respondents. As mentioned in the previous responses, sample size does not reflect reliability of data under the AHP approach, and while some respondents who completed Survey 1 chose not to complete Survey 2, CR values still met the needed criteria. One unexpected outcome related to response differences between Surveys 1 and 2 was our receipt of more conservation practitioner responses for Survey 2 than for Survey 1. This was unexpected because we only asked recruits to complete the second survey if they had already completed the first survey. This result likely arose because we had to exclude a small number of Survey 1 responses that were incomplete or contained response errors (7 surveys excluded for homeowners; 4 for conservation practitioners). Regardless of the explanation, we have no reason to believe that receiving slightly more Survey 2 results biased our results.

References:

Darko, A., Chan, A. P. C., Ameyaw, E. E., Owusu, E. K., Pärn, E., & Edwards, D. J. (2019). Review of application of analytic hierarchy process (AHP) in construction. International journal of construction management, 19(5), 436-452.

Dwivedi, P., & Alavalapati, J. R. (2009). Stakeholders’ perceptions on forest biomass-based bioenergy development in the southern US. Energy Policy, 37(5), 1999-2007.

Kurttila, M., Pesonen, M., Kangas, J., & Kajanus, M. (2000). Utilizing the analytic hierarchy process (AHP) in SWOT analysis—a hybrid method and its application to a forest-certification case. Forest policy and economics, 1(1), 41-52.

Margles, S. W., Masozera, M., Rugyerinyange, L., & Kaplin, B. A. (2010). Participatory planning: Using SWOT-AHP analysis in buffer zone management planning. Journal of sustainable forestry, 29(6-8), 613-637.

Masozera, M. K., Alavalapati, J. R., Jacobson, S. K., & Shrestha, R. K. (2006). Assessing the suitability of community-based management for the Nyungwe Forest Reserve, Rwanda. Forest Policy and Economics, 8(2), 206-216.

Shrestha, R.K., J.R. Alavalapati, and R.S. Kalmbacher. 2004. Exploring the potential for silvopasture adoption in south-central Florida: an application of SWOT–AHP method. Agricultural Systems 81(3):185-199.

Starr, M., Joshi, O., Will, R. E., & Zou, C. B. (2019). Perceptions regarding active management of the Cross-timbers forest resources of Oklahoma, Texas, and Kansas: A SWOT-ANP analysis. Land Use Policy, 81, 523-530.

---

## [Decision Letter · Decision Letter 1]

9 Dec 2021

PONE-D-21-17993R1Stakeholder perceptions of bird-window collisionsPLOS ONE

Dear Dr. Riggs,

Thank you for submitting your manuscript to PLOS ONE. After careful consideration, we feel that it has merit but does not fully meet PLOS ONE’s publication criteria as it currently stands. Therefore, we invite you to submit a revised version of the manuscript that addresses the points raised below during the review process.

We look forward to receiving your revised manuscript.

Kind regards,

Christopher A. Lepczyk

Academic Editor

PLOS ONE

Journal Requirements:

Additional Editor Comments (if provided):

Associate Editor:

Both reviewers found the revised manuscript to have addressed their comments. However, one reviewer has several remaining points that need to be addressed. In addition to these points a few minor items are as follows:

Abstract. Please include the main goal and objectives of the research in the Abstract.

Table 2. Indicate what bold terms mean. Also, table legend should be written as stand alone and explain more of what is in the table.

Acknowledgements. Suggest thanking the anonymous reviewers.

Reviewers' comments:

Reviewer's Responses to Questions

**Comments to the Author**

1. If the authors have adequately addressed your comments raised in a previous round of review and you feel that this manuscript is now acceptable for publication, you may indicate that here to bypass the “Comments to the Author” section, enter your conflict of interest statement in the “Confidential to Editor” section, and submit your "Accept" recommendation.

Reviewer #1: (No Response)

Reviewer #2: All comments have been addressed

2. Is the manuscript technically sound, and do the data support the conclusions?

Reviewer #1: (No Response)

Reviewer #2: Yes

3. Has the statistical analysis been performed appropriately and rigorously? 

Reviewer #1: (No Response)

Reviewer #2: Yes

4. Have the authors made all data underlying the findings in their manuscript fully available?

Reviewer #1: (No Response)

Reviewer #2: No

5. Is the manuscript presented in an intelligible fashion and written in standard English?

Reviewer #1: (No Response)

Reviewer #2: Yes

6. Review Comments to the Author

Reviewer #1: I’d like to thank the authors for their thorough and thoughtful revisions to the manuscript. I think the manuscript is just about ready for publication, but I do have some more minor suggestions below.

Abstract:

Not sure I understand the distinction between direct and indirect outcomes here- see comment in methods about this but I also think it needs to be clarified better here in the abstract: “respondents made pairwise comparisons between various strengths and weaknesses (direct outcomes related to collision management) and opportunities and threats

(indirect outcomes)”

Introduction

Line 100: I think there should be a list of citations here when discussing how SWOT and AHP are commonly used in human dimensions research

Methods

Line 122-124: I see the authors added this sentence to try to clarify the different components of SWOT, but I’m still confused: “Strengths and

122 weaknesses capture direct positive and negative outcomes, respectively, and therefore are

123 considered internal to the issue. Opportunities and threats respectively capture positive and

124 negative indirect outcomes.” What is a “direct outcome” and why are direct outcomes “internal”? What is an “indirect” outcome? I think more detailed explanation of these terms with examples would be helpful.

Line 151-152: awkward phrasing here: “they typically have public involvement…”

Line 163: What emails and social media platforms did you use to reach what types of stakeholders? More detail about this would be helpful so that the reader can better understand the types of biases in this sample (e.g., did you work with certain NGOs or other groups to post on their social media?). I also think its worth mentioning that this second group of people recruited through social media and email (as well as the gateway contacts) is a convenience sample- so basically, you used a combination of snowball sampling and convenience sampling. I also think a sentence or two here on why the convenience and snowball sampling approach (which is usually not sufficient for human dimensions research) is OK in the context of this study/your research questions.

Lines 185-186: Did you create the original list of SWOT factors based on a review of the literature? Talking with stakeholders? Some citations and explanation would be helpful.

Line 238: What is meant by “periodic reminders”? There are standard practices for when/how often to send reminders for surveys to increase response rate (see Dillman)- to what extent were these followed?

Line 307-311: I think the lack of data on who the respondents are (particularly with respect to geography) is a major limitation of this research. Collecting personally identifiable information is often done in HD research and is standard practice. I realized this can’t be changed at this point, but was there any information about respondents you could share here to give the reader a sense of who your sample is? Maybe at least providing demographic information about the gateway respondents? I know sample representativeness is brought up a bit in the discussion, but I think this limitation could be more strongly addressed here and in the discussion.

Discussion

Line 428-430: I think this finding can be more directly stated in the results; I had some trouble finding this exact result in the previous section.

Line 470: You mention this study by Menacho-Odio that also is a human dimensions study on bird-window collisions. This makes me think that the statement in the introduction that only one prior study on this topic has been done needs to be revised.

Line 512: I think it should be mentioned in this paragraph (or elsewhere in the discussion) the need for future research on the additional barriers and motivations to behavior change to design these programs. Some barriers/motivations were examined in this study (i.e. mostly beliefs), but many common factors influencing behavior change (e.g., social norms, behavioral control/efficacy, etc) were not. See the community based social marketing work by McKenzie-Mohr and Kollmuss and Agyeman 2002, Mind the gap: why do people act environmentally and what are the barriers to pro-environmental behavior?

Reviewer #2: (No Response)

7. PLOS authors have the option to publish the peer review history of their article (what does this mean?). If published, this will include your full peer review and any attached files.

Reviewer #1: No

Reviewer #2: No

---

## [Author Response · Author response to Decision Letter 1]

18 Jan 2022

Additional Editor Comments (if provided):

Associate Editor:

Both reviewers found the revised manuscript to have addressed their comments. However, one reviewer has several remaining points that need to be addressed. In addition to these points a few minor items are as follows:

Abstract. Please include the main goal and objectives of the research in the Abstract.

RESPONSE: We have revised accordingly to add the specific objectives of the research to the abstract (now found in Lines 29-32). We also revised the abstract to better present specific results and implications/conclusions that align with each of these 3 objectives (Lines 38-47).

Table 2. Indicate what bold terms mean. Also, table legend should be written as stand alone and explain more of what is in the table.

RESPONSE: We have revised the table legend (now in Lines 350-357) to describe the bold terms and to better elaborate on what the different values mean.

Acknowledgements. Suggest thanking the anonymous reviewers.

RESPONSE: Thank you for the suggestion. We have added thanks to the handling editor and anonymous reviewers in Lines 648-652.

Reviewers' comments:

Reviewer #1:

 I’d like to thank the authors for their thorough and thoughtful revisions to the manuscript. I think the manuscript is just about ready for publication, but I do have some more minor suggestions below.

RESPONSE: Thank you again for the positive and constructive comments that have helped us improve the manuscript.

Abstract

Not sure I understand the distinction between direct and indirect outcomes here- see comment in methods about this but I also think it needs to be clarified better here in the abstract: “respondents made pairwise comparisons between various strengths and weaknesses (direct outcomes related to collision management) and opportunities and threats (indirect outcomes)”

RESPONSE: Please see also our response to the later comment about this same issue in the Methods. Here in the abstract (Lines 34-38), we revised to add brief definitions and examples that should better clarify the definitions of strengths, weaknesses, opportunities, and threats.

Introduction

Line 100: I think there should be a list of citations here when discussing how SWOT and AHP are commonly used in human dimensions research

RESPONSE: We have added citations to relevant SWOT and AHP research here, and we restructured this sentence to make it clear that the citations support the approach of merging the two analyses. See Lines 108-109.

Methods

Line 122-124: I see the authors added this sentence to try to clarify the different components of SWOT, but I’m still confused: “Strengths and weaknesses capture direct positive and negative outcomes, respectively, and therefore are considered internal to the issue. Opportunities and threats respectively capture positive and negative indirect outcomes.” What is a “direct outcome” and why are direct outcomes “internal”? What is an “indirect” outcome? I think more detailed explanation of these terms with examples would be helpful.

RESPONSE: We understand that these terms might still cause some confusion, so we have added additional clarifying language, along with examples related specifically to the issue of bird-window collisions, to explain the differences more clearly among strengths, weaknesses, opportunities, and threats (see Lines 128-137). We hope that these additional changes prove clear enough for a broad readership. 

Line 151-152: awkward phrasing here: “they typically have public involvement…”

RESPONSE: We have clarified the wording here to read as (now at Lines 162-165): “they typically engage members of the public through activities such as…”

Line 163: What emails and social media platforms did you use to reach what types of stakeholders? More detail about this would be helpful so that the reader can better understand the types of biases in this sample (e.g., did you work with certain NGOs or other groups to post on their social media?). I also think its worth mentioning that this second group of people recruited through social media and email (as well as the gateway contacts) is a convenience sample- so basically, you used a combination of snowball sampling and convenience sampling. I also think a sentence or two here on why the convenience and snowball sampling approach (which is usually not sufficient for human dimensions research) is OK in the context of this study/your research questions.

RESPONSE: Regarding the comment about our use of email and social media platforms, we revised (Lines 174-187) to clarify who we contacted through email and how we did so, and about our approach of sharing recruitment materials (including a recruitment video on YouTube) through Facebook and Twitter. 

Regarding the comment about convenience sampling: although snowball sampling has been periodically used as a purposive sampling tool to gauge stakeholder perceptions in the conservation social science literature (e.g. Karanth et al. 2008, Said et al. 2016, Krupa et al, 2018), the reviewer is correct that random sampling frameworks have historically dominated mainstream human dimensions research. Concerning our study, it is important to note that the research goal was to depict how similar (or different) the key stakeholder groups were regarding their opinions on bird-window collisions. During the research design stage, we framed our study following the typical protocols of the SWOT-AHP platform, which often uses purposive data collection tools such as in-person focus groups (e.g. Dwivedi and Alavalapati 2009) or mixed-mode sampling with in-person meetings for some stakeholders and web surveys for others (e.g. Starr et al. 2019). However, since we did not have a comprehensive list of stakeholders, we utilized snowball sampling, gateway contacts, and targeting of select HOAs in Stillwater to recruit survey participants. Nonetheless, as already described in Lines 332-337 of the revised manuscript, the range of consistency ratios (CR) for our response data is sufficient to discern the internal consistency of stakeholder opinions. Our study results should therefore broadly represent consensus opinions of each stakeholder group. To address this reviewer comment, we revised the limitations section of the Discussion (Lines 566-568) to acknowledge that our convenience-based sampling contributed to the issue of representativeness of respondents (which we had already discussed in relation to limited sample size). Later in the same paragraph we also had described how consistency ratios indicate that our results should be reliable despite these issues related to convenience sampling and representativeness.

References:

Said, A., Tzanopoulos, J., & MacMillan, D. (2016). Bluefin tuna fishery policy in Malta: the plight of artisanal fishermen caught in the capitalist net. Marine Policy, 73, 27-34.

Karanth, K. K., Kramer, R. A., Qian, S. S., & Christensen Jr, N. L. (2008). Examining conservation attitudes, perspectives, and challenges in India. Biological Conservation, 141(9), 2357-2367.

Krupa, M., Cenek, M., Powell, J., & Trammell, E. J. (2018). Mapping the stakeholders: Using social network analysis to increase the legitimacy and transparency of participatory scenario planning. Society & Natural Resources, 31(1), 136-141.

Dwivedi, P., & Alavalapati, J. R. (2009). Stakeholders’ perceptions on forest biomass-based bioenergy development in the southern US. Energy Policy, 37(5), 1999-2007.

Starr, M., Joshi, O., Will, R. E., & Zou, C. B. (2019). Perceptions regarding active management of the Cross-timbers forest resources of Oklahoma, Texas, and Kansas: A SWOT-ANP analysis. Land Use Policy, 81, 523-530.

Lines 185-186: Did you create the original list of SWOT factors based on a review of the literature? Talking with stakeholders? Some citations and explanation would be helpful.

RESPONSE: We revised the manuscript to clarify (now in Lines 203-206) that we developed this initial list of candidate SWOT factors based on our extensive subject matter expertise, which includes familiarity with the scientific and gray literature on this topic, and our years of interactions and collaborations with key stakeholders in federal and state agencies and non-profit conservation organizations. As already stated in the manuscript, we also sought feedback from 3 external experts to create the final list of SWOT factors used in this study.

Line 238: What is meant by “periodic reminders”? There are standard practices for when/how often to send reminders for surveys to increase response rate (see Dillman)- to what extent were these followed?

RESPONSE: We revised to clarify (now in Lines 258-260) that, for all stakeholder groups and sampling approaches, we sent one reminder to complete the survey two weeks after the original participation request (we also added a citation to the Dillman reference supporting this follow-up method).

Line 307-311: I think the lack of data on who the respondents are (particularly with respect to geography) is a major limitation of this research. Collecting personally identifiable information is often done in HD research and is standard practice. I realized this can’t be changed at this point, but was there any information about respondents you could share here to give the reader a sense of who your sample is? Maybe at least providing demographic information about the gateway respondents? I know sample representativeness is brought up a bit in the discussion, but I think this limitation could be more strongly addressed here and in the discussion.

RESPONSE: Thanks for the opportunity to respond to and clarify this issue. Following well-documented AHP protocols, several researchers have used a very small number of expert opinions to identify stakeholder perceptions regarding natural resource issues. For example, Shrestha et al. (2004) used opinions of three leaders, a research/extension specialist (to represent agriculture research/extension professionals) and two ranchers (to represent landowners), to explore the potential for silvopasture adoption in south-central Florida (page 190, Shrestha et al. 2004). Similarly, Kurtitila et al. (2000) used opinions of two experts to compare strengths, weaknesses, opportunities, and threats associated with forest certification in eastern Finland (page 47, Kurtitila et al. 2000). Masozera et al. (2006) used a total of 11 participants representing three groups (5 from one group and 3 from the other two groups) in their effort to assess suitability of community-based management for the Nyungwe Forest Reserve in Rwanda. In other words, a small sample size is acceptable as long as it meets the consistency ratio (CR) criterion. In these typical studies, researchers do not report personally identifiable information as the sample size is extremely low. Although the sample size for our study (n=52 for the first survey and n=33 for the second survey) is larger than the above-highlighted studies, we unfortunately did not have a pre-established contact list for our respondents and we did not know how many respondents would actually fill out the survey. Therefore, we did not ask for any personally identifiable information. In retrospect, our readers would benefit from this information, and we recognize this lack of information as a limitation. To address this comment, we have added acknowledgment of this limitation in the Discussion (Lines 594-598). Specifically, we note that we did not collect demographic information from respondents, nor did we know the geographic representation of our sample other than for gateway contacts. We then highlight that future research could evaluate how perceptions about bird-window collisions vary regionally and in relation to various demographic factors, since the factors that influence behaviors, perceptions, and conflicts can vary regionally and among demographic groups (e.g., among different age groups).

References:

Kurttila, M., Pesonen, M., Kangas, J., & Kajanus, M. (2000). Utilizing the analytic hierarchy process (AHP) in SWOT analysis—a hybrid method and its application to a forest-certification case. Forest policy and economics, 1(1), 41-52.

Masozera, M. K., Alavalapati, J. R., Jacobson, S. K., & Shrestha, R. K. (2006). Assessing the suitability of community-based management for the Nyungwe Forest Reserve, Rwanda. Forest Policy and Economics, 8(2), 206-216.

Shrestha, R. K., Alavalapati, J. R., & Kalmbacher, R. S. (2004). Exploring the potential for silvopasture adoption in south-central Florida: an application of SWOT–AHP method. Agricultural Systems, 81(3), 185-199. 

Discussion

Line 428-430: I think this finding can be more directly stated in the results; I had some trouble finding this exact result in the previous section.

RESPONSE: Thank you for this suggestion. Although the values supporting this finding were already presented in Lines 339-341 of the revised manuscript, we agree that the comparison could be made more directly, so we added an additional line directly comparing the group priority values for threats between homeowners and conservation practitioners (Lines 345-346).

Line 470: You mention this study by Menacho-Odio that also is a human dimensions study on bird-window collisions. This makes me think that the statement in the introduction that only one prior study on this topic has been done needs to be revised.

RESPONSE: Thank you for catching this. We revised this paragraph to note that there have only been two studies evaluating human dimensions of the bird-window collision issue, and we added an additional sentence to highlight the Menachio-Odio study (Line 91-93).

Line 512: I think it should be mentioned in this paragraph (or elsewhere in the discussion) the need for future research on the additional barriers and motivations to behavior change to design these programs. Some barriers/motivations were examined in this study (i.e. mostly beliefs), but many common factors influencing behavior change (e.g., social norms, behavioral control/efficacy, etc) were not. See the community based social marketing work by McKenzie-Mohr and Kollmuss and Agyeman 2002, Mind the gap: why do people act environmentally and what are the barriers to pro-environmental behavior?

RESPONSE: We followed the reviewer’s suggestion and revised to add a mention of the need for this type of future research (along with citation to the Kollmuss and Agyeman reference). However, instead of adding this text in the location referenced by this comment, we added it to the later Discussion paragraph that describes several important research needs related to the human dimensions of bird-window collisions (see Lines 584-593).

Reviewer #2: (No Response)

---

## [Editor Report · Decision Letter 2]

20 Jan 2022

Stakeholder perceptions of bird-window collisions

PONE-D-21-17993R2

Dear Dr. Riggs,

We’re pleased to inform you that your manuscript has been judged scientifically suitable for publication and will be formally accepted for publication once it meets all outstanding technical requirements.

Kind regards,

Christopher A. Lepczyk

Academic Editor

PLOS ONE
---

## [Editor Report · Acceptance letter]

2 Feb 2022

PONE-D-21-17993R2 

Stakeholder perceptions of bird-window collisions 

Dear Dr. Riggs:

I'm pleased to inform you that your manuscript has been deemed suitable for publication in PLOS ONE. Congratulations! Your manuscript is now with our production department. 

Kind regards, 

on behalf of

Dr. Christopher A. Lepczyk 

Academic Editor

PLOS ONE